# Uncertainty-driven Sanity Check: Application to Postoperative Brain Tumor Cavity Segmentation

**Alain Jungo**[*]
ISTB
University of Bern
`alain.jungo@istb.unibe.ch`

**Raphael Meier**[†]
SCAN
Bern University Hospital

**Ekin Ermis**[‡]
University Clinic for Radio-oncology
Bern University Hospital

**Evelyn Herrmann**[‡]
University Clinic for Radio-oncology
Bern University Hospital

**Mauricio Reyes**[*]
ISTB
University of Bern

## Abstract

Uncertainty estimates of modern neuronal networks provide additional information next to the computed predictions and are thus expected to improve the understanding of the underlying model. Reliable uncertainties are particularly interesting for safety-critical computer-assisted applications in medicine, e.g., neurosurgical interventions and radiotherapy planning. We propose an uncertainty-driven sanity check for the identification of segmentation results that need particular expert review. Our method uses a fully-convolutional neural network and computes uncertainty estimates by the principle of Monte Carlo dropout. We evaluate the performance of the proposed method on a clinical dataset with 30 postoperative brain tumor images. The method can segment the highly inhomogeneous resection cavities accurately (Dice coefficients $0.792 \pm 0.154$). Furthermore, the proposed sanity check is able to detect the worst segmentation and three out of the four outliers. The results highlight the potential of using the additional information from the model's parameter uncertainty to validate the segmentation performance of a deep learning model.

## 1 Introduction

Recent advances in supervised machine learning enable human level performances [12, 15] of fully-automated medical imaging systems, but modern deep learning models still lack in robustness when applied to real-world clinical data. The considerable variability in real-world data can lead to significant mispredictions. Such errors hinder the utilization of fully-automated systems in clinical practice, where life-critical decisions are made. Uncertainty estimates can increase the understanding of the underlying model and help to foster acceptance of deep learning models by

---

[*]Institute for Surgical Technology and Biomechanics (ISTB), University of Bern, Bern, Switzerland

[†]Support Center for Advanced Neuroimaging (SCAN), Institute for Diagnostic and Interventional Neuroradiology, Inselspital, Bern University Hospital, University of Bern, Bern, Switzerland

[‡]University Clinic for Radio-oncology, Inselspital, Bern University Hospital, University of Bern, Bern, Switzerland

1st Conference on Medical Imaging with Deep Learning (MIDL 2018), Amsterdam, The Netherlands.

providing additional information about the predictions. This additional information enables building systems with improved user-feedback or automatic detection of challenging cases, necessitating human monitoring. Such systems add a layer of quality assurance, and hence makes them better suited for clinical practice.

Various works have been proposed to produce uncertainty estimates in neural networks. Most of them build on Bayesian neural networks (BNN) [11, 13] which assign prior distributions on the network weights. Since exact inference is intractable in BNN's, different approximations have been presented [4, 1, 3]. Monte Carlo dropout (MC dropout) proposed by Gal & Ghahramani [3] is probably the most popular approximation method. Its definition permits a simple and straightforward realization in most modern neural networks. Other approaches make use of generative adversarial networks (GANs) to produce uncertainty estimates [17, 14]. Recently, Lakshminarayanan et al. proposed a non-Bayesian way of calculating uncertainty from ensembles [10]. Despite the recent advances in Bayesian deep learning, less work focuses on the computation of pixel/voxel-wise uncertainty maps. Existing work includes uncertainty estimates for road scene segmentations [7, 8] and applications in image quality transfer for medical images [16]. However, these approaches do not make further use of the computed uncertainties.

We propose a convolution neural network (CNN) for postoperative brain tumor cavity segmentation and employ the model's uncertainty to identify challenging cases necessitating expert review. In radiotherapy, postoperative cavities and the residual tumor are manually segmented to determine the broader clinical target volume to be irradiated. This time-consuming task could greatly benefit from intelligent automation with error detection. Therefore, our contribution is twofold: (a) we propose a simple sanity check based on the model's segmentation uncertainty and (b) we present the first CNN approach for the delineation of postoperative brain tumor cavities, to the best of our knowledge.

## 2 Material and Methods

### 2.1 Data

**Clinical dataset.** Postoperative magnetic resonance images with 30 subjects to evaluate the post-operative (i.e., after tumor resection) status of glioblastoma patients are available. The images were acquired in the four standard sequences (T1-weighted (T1), T1-weighted post-contrast (T1c), T2-weighted (T2), and Fluid-attenuated inversion-recovery (FLAIR)). All sequences of a subject are co-registered and anonymized, i.e., skull-stripped. Three clinical radiation oncology experts with different levels of expertise (two years, four years, and over six years of clinical experience) created the label maps. These maps delineate the cavity after tumor resection and are used for radiotherapy planning. The difficulty of the task is associated with the presence of blood products, cerebrospinal fluid infiltration and air pockets, which change the appearance of the resection cavity. The dataset ground truth was created by a majority voting of the three expert segmentation. Figure 1 illustrates the four sequences and the ground truth in the dataset on an exemplary image slice.

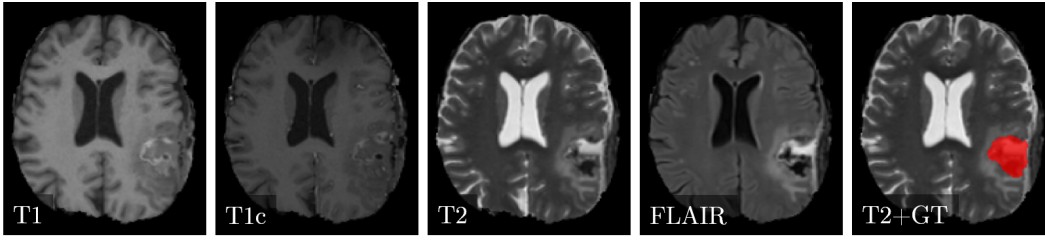

Figure 1: Example case of the postoperative brain tumor dataset. From left to right the images correspond to the four sequences T1-weighted (T1), T1-weighted post-contrast (T1c), T2-weighted and Fluid-attenuated inversion-recovery (FLAIR). The image on the right shows the ground truth (GT) as an overlay on the T2 image.

**Preprocessing.** We apply two preprocessing steps before feeding the data in the neural network. The first step is a rigid registration of the subject to an atlas. The registration is performed for the T1c sequence and then applied to the three other sequences (i.e. T1, T2, and FLAIR). Due to the

translation invariance of convolutional neural networks, this inter-subject registration is not exigent but enables transforming all images and sequences to unit-voxel size ($1 \times 1 \times 1$mm) with identical dimensions ($200 \times 200 \times 200$). The second preprocessing step is a z-score intensity normalization ($\mu = 0$, $\sigma = 1$). It is performed for each subject and sequence individually.

## 2.2 Architecture

**Fully-convolutional DenseNet.** We use a fully-convolutional DenseNet architecture [5], which is an adaptation of DenseNet [6] for semantic segmentation tasks. The architecture has a pooling/upsampling structure with skip connections that help to capture context and local features simultaneously. We apply four levels of pooling/upsampling. Each level contains a dense block of four dense block layers (i.e., batch normalization, ReLU, convolution, dropout). Table 1 lists the architecture details. If not otherwise mentioned, every convolution has a kernel size of $3 \times 3$ and is followed by a dropout with $p = 0.2$.

The network requires two-dimensional slices of all four image sequences (i.e. T1c, T1, T2, and FLAIR) as input. Be $I \in \mathbb{R}^{m \times n \times 4}$ the four-channeled input with in-plane resolution $m = 200$ and $n = 200$. The output is a probability distribution $p(Y \mid I)$ with $Y \in \mathcal{C}^{m \times n}$ where $\mathcal{C} = \{0, 1\}$ is the set of classes, i.e. background and cavity. We note that although the prediction is performed slice-wise, the formulation can be extended to volumes with $\mathbf{I} \in \mathbb{R}^{l \times m \times n \times 4}$ and $\mathbf{Y} \in \mathcal{C}^{l \times m \times n}$ with $l$ being the slices.

Table 1: Description of the architecture used for the cavity segmentation task. A dense block is a densely connected building block of the DenseNet architecture [5]. Transition down (TD) consists of one dense block layer (i.e., batch normalization, ReLU, convolution, dropout) and an additional max pooling. Transition up (TU) represents the upsampling convolution [6]. The skip connections between dense blocks of equal pooling/upsampling level are not shown.

| Building Block | Channels | Size |
|---|---|---|
| Input | 4 | $200 \times 200$ |
| Convolution & Dropout | 48 | $200 \times 200$ |
| Dense Block (4 layers) & TD | 96 | $200 \times 200$ |
| Dense Block (4 layers) & TD | 144 | $100 \times 100$ |
| Dense Block (4 layers) & TD | 192 | $50 \times 50$ |
| Dense Block (4 layers) & TD | 240 | $25 \times 25$ |
| Dense Block (4 layers) | 288 | $12 \times 12$ |
| TU & Dense Block (4 layers) | 336 | $25 \times 25$ |
| TU & Dense Block (4 layers) | 288 | $50 \times 50$ |
| TU & Dense Block (4 layers) | 240 | $100 \times 100$ |
| TU & Dense Block (4 layers) | 192 | $200 \times 200$ |
| $1 \times 1$ Convolution | 2 | $200 \times 200$ |
| Softmax | 2 | $200 \times 200$ |

**Oriented slice-wise prediction.** The unprocessed image volumes have an anisotropic resolution depending on the subject and sequence. Due to this, we favor a slice-wise (i.e., two-dimensional) over volume-wise (i.e., three-dimensional) approach. The benefit of working with slices is twofold: (a) no patching due to memory limitations is required and (b) significantly faster convolutions. In contrast, the volume information is lost. To alleviate for this loss, we determine the volume predictions $p(\mathbf{Y} \mid \mathbf{I})$ in all three anatomical planes (i.e. axial ($a$), coronal ($c$), sagittal($s$)) and calculate the final predictions as the average $1/3 \sum_{j \in \{a,c,s\}} p(\mathbf{Y}_j \mid \mathbf{I}_j)$.

**Training and model selection.** We use Adam [9] optimizer with a learning rate of $10^{-4}$ to optimize the cross-entropy loss. After 20 epochs of training with batch size 16, we select the last model for testing. Training takes approximately four hours on an NVIDIA Titan Xp GPU. The test time for one subject volume is approximately 10 seconds.

## 2.3 Uncertainty

**Uncertainty estimation.** Dropout regularization can be interpreted as an approximation for Bayesian inference over the weights of the network [3]. In contrast to the traditional weight scaling, the dropout is applied during training and test time. At test time, the dropouts produce randomly sampled networks, which can be viewed as Monte Carlo samples over the posterior distribution $p(\mathbf{W} \mid \mathcal{I}, \mathcal{Y})$ of the model weights $\mathbf{W}$ with $\mathcal{I}$ and $\mathcal{Y}$ the set of images and labels in the dataset. $T$ network samples are used to produce one prediction with uncertainty estimation. The classification of one voxel is determined by the average of posterior probabilities

$$p(y \mid I) = \sum_{c \in \mathcal{C}} \left( \frac{1}{T} \sum_{t=1}^{T} p(y_t = c \mid I) \right),$$

over $T$ predictions. The class uncertainty for one voxel can be computed with the approximated predictive entropy [2]

$$h \approx - \sum_{c \in \mathcal{C}} \left( \frac{1}{T} \sum_{t=1}^{T} p(y_t = c \mid I) \right) \log \left( \frac{1}{T} \sum_{t=1}^{T} p(y_t = c \mid I) \right).$$

We use a $T$ of 20 for the experiments.

**Uncertainty-driven sanity check.** Uncertainty estimates are useful because they provide additional information on the network's confidence. However, it is not feasible for a clinician to go through every predicted case, and corresponding uncertainty map, to sort out the failing cases. Therefore, we propose to automatically identify challenging cases for further assessment based on the calculated uncertainties. To do this, we extract the essential information contained in the uncertainty maps, and summarize into a *doubt* score. The *doubt* score $dbt$ for one predicted image is defined as

$$dbt = \sum_{i=1}^{N} \mathbb{1}\{k_i = 1\} w_i h_i,$$

where $N$ is the voxel count and $h_i$ the entropy at voxel $i$. The result is a sum of the uncertainty weighted by a distance map $w$ and masked by a binary image $k$.

To produce the mask image $k$, we (a) dilate the outline of the segmentation result twice, (b) invert the mask such that everything except the dilated outline is one, and (c) restrict the mask to the regions where the uncertainty exceeds 0.5. By applying this mask, we reduce the typically high uncertainty at the class boundaries. Due to its consistent presence, this boundary uncertainty does not contribute to the detection of challenging cases.

For radiotherapy planning, the negative impact of false positives increases with the distance to the cavity. Therefore, the uncertainty is weighted by a distance map that we compute by (a) determining the outline of the segmentation result, and (b) determining the Euclidean distance from every voxel to the outline.

## 2.4 Evaluation

We evaluated the segmentation performance and the uncertainty-driven sanity check separately. To analyze the segmentation performance we used the Dice coefficient, Hausdorff distance (95[th] percentile) and the volume similarity. Additionally, we qualitatively examined the best and worst performing cases with respect to the Dice coefficient. The sanity check was evaluated on its ability to sort out the worst segmentation cases by thresholding the *doubt* score.

We performed all the experiments on a six-fold cross-validation, i.e., 25 training images and five testing images. Furthermore, since we observed slightly improved segmentation performances with weight scaling, we only used the MC dropouts for uncertainty estimates.

# 3 Results

First, we present the result of the segmentation performance before we focus on the uncertainty. For the sake of clarity, we use subject identifiers (i.e., sample number) for the presented cases throughout the section.

## 3.1 Segmentation

Figure 2 illustrates the quantitative results of the segmentation performance on the 30 cases of the dataset. Every dot corresponds to a sample. The ones marked with numbers are selected according to their achieved Dice score. The selection contains the two best (samples 21 and 22) and the three worst performing cases (samples 10, 24, 2). Table 2 shows a statistical summary of the results. The proposed method achieved a median Dice of 0.839. This is close to the variability in the manual expert delineations, which has a median Dice coefficient of 0.85.

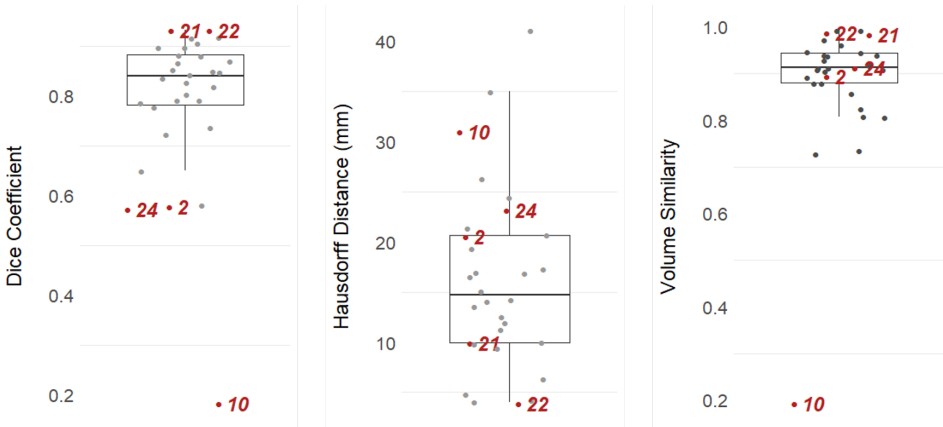

Figure 2: Quantitative results of the segmentation performance. Numbers indicate the sample number of selected examples based on their Dice score.

Table 2: Summarized results of the proposed network on the clinical postoperative brain tumor cavity dataset (Avg: average, Std: standard deviation, Med: median).

| Metric | Avg | Std | Med |
|---|---|---|---|
| *Dice coefficient* | 0.792 | 0.154 | 0.839 |
| *Hausdorff distance (mm)* | 16.24 | 9.07 | 14.74 |
| *Volume similarity* | 0.881 | 0.146 | 0.913 |

Figure 3 visualizes the segmentation results for the selected samples. The first and second row correspond to the best-performing cases. Their segmentation is close to ground truth and the uncertainty is mainly present at the boundary between foreground (i.e., cavity) and background. In contrast, samples 2, 24, and 10 show the worst performances. The network is struggling with the inhomogeneous cavity of sample 2. There is no clear delineation in the input sequences, which makes the task difficult. Sample 24 is most probably flawed because of the erroneous registration (compare T1 and T2 in Figure 3). Sample 10 is failing although the cavity is clearly visible in the T1 and T2 sequence. Further analysis of this case pointed to a particular hypointense cavity in the FLAIR sequence which appears isointense to cerebrospinal fluid and might confuse the network. Additionally, all three cases yielded increased uncertainties.

## 3.2 Uncertainty-driven sanity check

Figure 4 shows the relation between the *doubt* score and Dice coefficient for the 30 samples of the dataset. The orange horizontal line represents a possible threshold value for the sanity check (e.g. pre-defined by the expert). Every case above the threshold would be flagged as particular

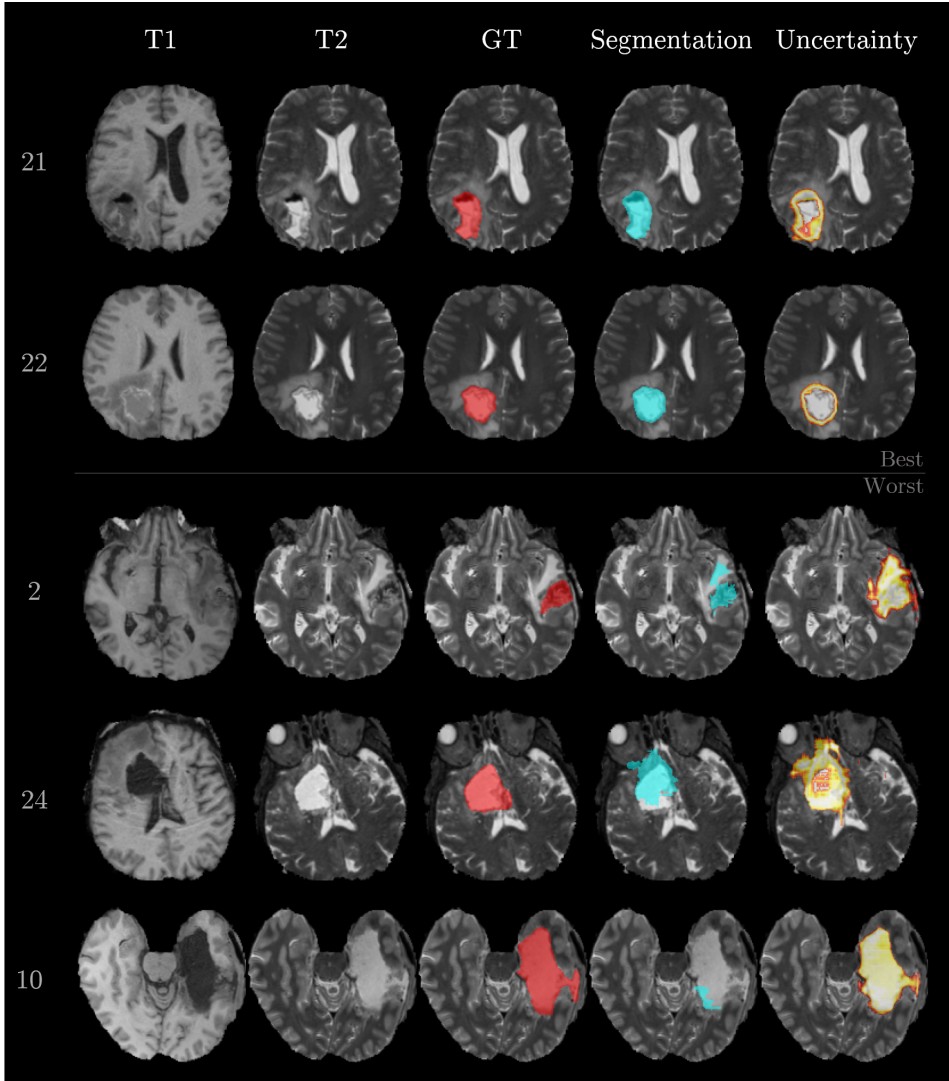

Figure 3: Exemplary cases of segmented postoperative brain tumor cavities. The row numbers on the left correspond to the sample number. The two top rows show the best performing cases and the three bottom rows the worst cases. Columns represent the type of information (input sequences T1 and T2, ground truth (GT), and the network outputs, i.e., segmentation and uncertainty). The input sequences T1c and FLAIR are omitted.

error-prone and would require specific review by a clinician. We add another threshold (0.75) on the Dice coefficient (dashed green vertical line in Figure 4) to separate the good segmentation from the ones that would require improvement. In terms of uncertainty, both thresholds together form regions of true positives (upper left quadrant), false positives (upper right quadrant), false negatives (lower left quadrant) and true negatives (lower right quadrant). The majority of the samples (22 out of 30) are situated either in true positives (4 samples) or the true negatives (18 samples).

In addition to the qualitative examples in Figure 3, Figure 5 focuses on examples situated in the false positives quadrant (sample 3 and 17) and false negatives quadrant (sample 17). The massive brain shift of sample 3 (top row in Figure 5) does not cause the segmentation to fail but is resulting in an increased uncertainty. Likewise, the uncertainty is increased at the meninges and subarachnoid space for sample 27. Both cases lead to large *doubt* scores because of the high uncertainties located far away from the segmented cavity. In contrast, sample 17 does mainly show uncertainty around the erroneous segmentation which leads to a small *doubt* score.

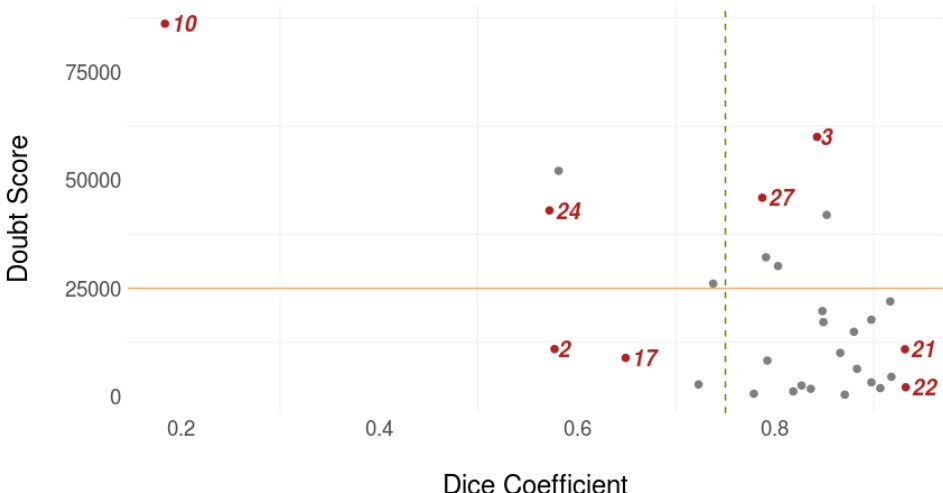

Figure 4: Relation between the *doubt* score and Dice coefficient. The orange horizontal line represents a possible threshold value for the sanity check, and the dashed green vertical line corresponds to a threshold for good segmentation performance. Together they form quadrants of true positives (upper left), false positives (upper right), false negatives (lower left) and true negatives (lower right).

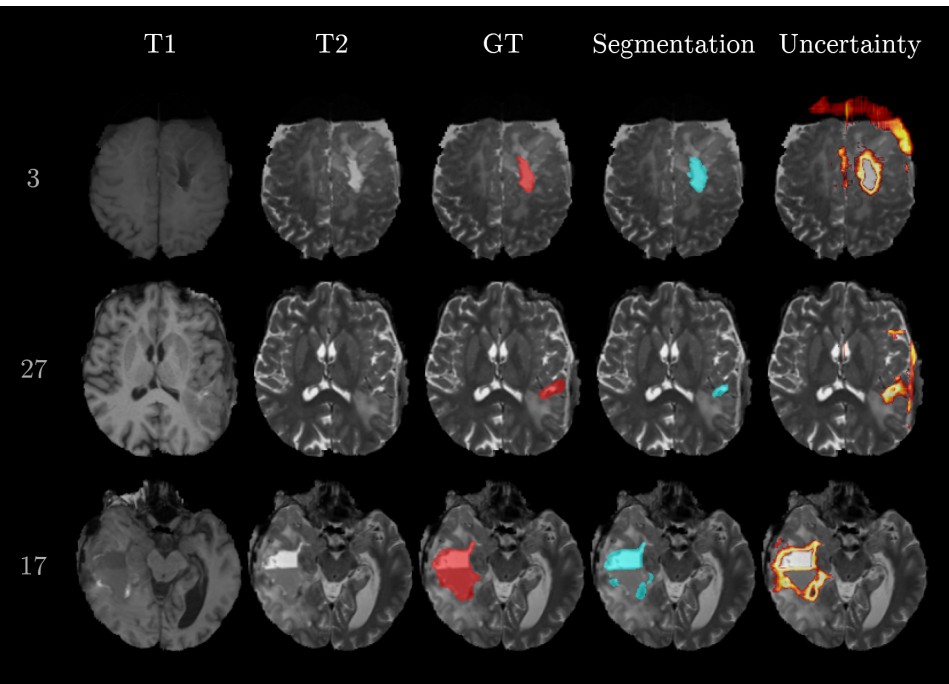

Figure 5: Examples of segmented postoperative brain tumor cavities. The row numbers on the left correspond to the sample number. The two top rows correspond to samples with high *doubt* scores and Dice coefficients (false positives). The bottom row corresponds to a sample with low *doubt* score and Dice coefficient (false negative).

## 4   Discussion

The results show that our proposed method produces accurate segmentation of postoperative brain tumor cavities. Furthermore, the experiments highlight the potential of computing the model's uncertainty next to the segmentation. Typically, one would expect a rather low *doubt* scores for good segmentation results, which would allow a simple detection of the failing segmentation cases.

But the results revealed cases with high Dice coefficient and high *doubt* scores. Even though such cases were classified as false positives in terms of failure detection, their claim for review is justified. Additionally, the results show the importance of the spatial prior introduced in the *doubt* score as indicator for the identification of challenging cases. This suggests that a transfer of the presented method to another segmentation task is possible but might require adaptations on the spatial prior.

In radiotherapy planning and medical imaging in general, patient safety is the priority. Hence, introducing sanity checks in automated segmentation tools may be a step towards a compromise between full supervision by the clinician and full automation. By adding parameters permitting the control of the degree of automation, the level of supervision could be configured according to the seriousness of a task. High throughput analysis (e.g., clinical trials or drug-assessment) could greatly benefit from such a controlled automation.

Although the presented cavity segmentation approach achieves good results, it still contains few outliers. In radiotherapy planning, outliers can impact target region for irradiation. Therefore, it is essential to reduce the wrongly segmented foreground as much as possible. Improving the sanity check for small regions of wrongly segmented foreground would be of benefit. For now, the proposed sanity check focuses on segmentation uncertainty. To further enhance the detection (reduce false positives and false negatives) other sources of information may also be considered. Finally, with more parameters, the sanity check could itself be a neural network that is trained to detect segmentation failures.

## 5   Conclusion

We presented a fully-convolutional neural network for postoperative brain tumor cavity segmentation with an uncertainty-driven sanity check. This check encodes a spatial prior and existing knowledge about Bayesian uncertainty estimation to automatically detect segmentation results that require expert review.

**Acknowledgments**

This work was supported by the Swiss National Foundation, grant number 169607.

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
