# OpenReview forum: "Uncertainty-driven Sanity Check: Application to Postoperative Brain Tumor Cavity Segmentation"
_MIDL.amsterdam/2018/Conference — MIDL 2018 Poster_

### Review · AnonReviewer2 · 2018-05-03
**The authors propose a fairly simple approach for unsupervised quality control of deep neural network segmentations which seems overall convincing as a first step towards auto quality assurance. A weakness is the relatively small data set size of 30 patients only (significance of results?). The spatial prior in the quality score is not yet convincing. It would be interesting to see future evaluations on larger data sets and modifications to the spatial prior.**

**Rating:** 4
**Confidence:** 2

**Review:**

The work uses a fully convolutional DenseNet for brain tumor cavity segmentation and proposes to use a doubt score for uncertainty estimation.The score is derived from network uncertainty using the Monte Carlo dropout method. The derived sanity check is supposed to help understanding of neural network outputs and to add to their quality assurance by automatic identification of segmentation errors which need expert review.

Pros:

- Automatic (and unsupervised) quality assurance of fully-automated segmentation methods can be of high significance, especially when large datasets need to be processed.

- The proposed method does not need a reference segmentation for computing the "doubt score" which makes it possible to apply also to unseen data without reference segmentation available, in contrast to other metrics such as the Dice score. Therefore it is also independent from a human observer.

- The proposed method for the sanity check is fairly simple and could also be applied to a variety of different network architectures, not just the one used in the paper.

- Reference data quality: the authors use contours for training and evaluation produced through a majority vote of 3 clinical experts which is likely to be of higher quality than data from a single observer and will lead to more robust quantitative results.

- Overall, the paper is clearly written and has a logic structure which is easy to follow.

Cons:

- Size of the dataset: A main motivation by the authors is that "deep learning methods still lack robustness when applied to real-world clinical data" which can be subject to "considerable variability" and thus lead to "significant mispredictions". However, they evaluate their sanity check method on a rather small dataset of 30 patients only, which will most probably not capture most of the variability that could be expected (also during training). Also, variability in MR images is often due to use of different scanners and different protocols in different clinics, not just anatomical variability. This is not adressed and it is not explicitely stated whether all 30 patients were imaged in the same clinic and scanner.

- With a sensitivity of 4/7 and a false discovery rate of 5/9, the doubt score as a means of identifying bad segmentations (with a Dice <0.75) is in its current form not completely convincing. The specificity is higher, but it seemed that the main motivation was to detect the bad segmentations with a high confidence for expert review. Then again, the dataset used is probably too small to draw too many conclusions from these numbers.

- The "importance of the spatial prior" is not completely evident from the results. In some cases, the distance weights in the doubt score seem to introduce false positives (e.g. case 3: high doubt score due to far off region with high uncertainty, even though the segmentation quality seems acceptable) and false negatives (e.g. case 17: on the shown slice about half of the cavity is missing, but as this is close to the boundary, the doubt score is still low). It would be interesting to see results such as in Figure 4 when omitting the distance weight or using a non-linear transform of the euclidean distance.

Other comments:

- It is unclear how the number of Monte Carlo dropout networks was chosen to be T=20 and how much this number impacts the resulting doubt score. Is there some sort of convergence as T grows? The choice of threshold for the doubt score seems also not very intuitive.

- The need for registration to an atlas is not obvious, resampling and cropping could be done without. Is there any other reason to use the atlas?

- The authors state that a 2D approach is favored over 3D also because of the anisotropic resolution in the unprocessed images. This is not very convincing as resampling to isotropic voxels is done during pre-processing before the network sees any images.

- The notation "dense layer" to refer to a sequence of BatchNorm, Conv, ReLU etc. within a "dense block" might be a bit confusing as "dense layer" often refers to fully connected layers.

- In the last section of the discussion the notations false/true positives/negatives are used to refer to the segmentation result. This is a bit confusing, as before these terms were used to describe the sanity check results. Maybe rephrase that last part?

- It could be nice to see the doubt scores in the figures 3 and 5 for direct comparison.

**Special Issue:**

Yes

---

> ### Comment · ~Alain_Jungo1 · 2018-05-14
> **Response to Reviewer2**
>
> We thank you for your positive comments about our manuscript. Please find below our clarification to the comments.
>
> We clarify that all 30 images stem from the same clinic, but different scanners were used.
>
> The motivation of the spatial prior is clinical-driven. Errors close to actual segmentation are less critical for usages such as in radiotherapy planning than uncertain regions far from the segmentation boundary. We thus consider false positives that reveal problematic regions (as in case 17) as helpful and not as a failure. During development, we also compared to the setup without spatial prior, but it resulted in doubt values that were driven mainly by minor regions close to the segmentation boundary.
>
> The number of Monte Carlo samples we chose is a tradeoff between the benefit of many samples and the computational requirement (each sample requires a forward pass in the network).
>
> We agree that the atlas registration is not required. However, it simplifies the cropping to a minimal mutual size among the subjects.
>
> We favor a slice-wise approach over a volume-wise (3D) approach due to the difference between in-plane resolution (typically around 1x1mm2) and inter-slice spacing (up to 6mm) of the unprocessed images (as recently shown in Baumgartner et al., 2017. An Exploration of 2D and 3D Deep Learning Techniques for Cardiac MR Image Segmentation. https://link.springer.com/chapter/10.1007/978-3-319-75541-0_12). Although resampled to unit voxel resolution, we expect that the network obtains the segmentation crispiness from the good in-plane resolution, which motivates us to process the in-plane separately from the out-planes. Moreover, the 2D approach is fast to compute. Computation times gain in importance when performing Monte Carlo sampling for the uncertainty estimates.
>
> We agree that the notation “dense layer” is misleading. We will rename it to “dense block layer” (as in [6]) in case of acceptance.  Likewise, we will rephrase the last section of the discussion to avoid confusion with the terms used before.

---

### Review · AnonReviewer1 · 2018-05-09
**A well constructed paper describing application of deep learning methodologies to brain tumor cavity segmentation.**

**Rating:** 3
**Confidence:** 2

**Review:**

The paper describes a deep neural network applied to post-operative MR images for the segmentation of brain tumor cavities.  In addition it estimates the certainty of the segmentation result across the image to potentially identify images/segmentations which would require manual checking in a clinical environment.  The paper is well written and methods are clearly described.  The dataset is appropriately split into training/test using crossfold validation, but no validation set is used to set network architecture and parameters.  This implies that the network is tuned to obtain optimal results on the test set used here and may not generalise with similar performance.
The methods described (network for segmentation and dropout for uncertainty estimation) are not novel in themselves, only the application.  The segmentation results on a dataset of 30 subjects are good and it is demonstrated that the uncertainty estimate is useful (if not foolproof) in identifying possible incorrect segmentations for manual checking.
While I would not reject the paper immediately I would give only a tentative recommendation to accept based on the low novelty value of the methodology.

**Special Issue:**

No

---

> ### Comment · ~Alain_Jungo1 · 2018-05-14
> **Response to Reviewer1**
>
> We thank you for your comments about our manuscript. Please find below our clarification to the comments.
>
> While it is true that through cross-validation the network is tuned on the test set, we tried to keep the bias low by defining all hyperparameters only on one split (out of the 6) of the cross-validation. The remaining five (training-test) splits were trained with the exact same hyperparameters and an equal stopping criterion.
>
> We clarify that we do not claim novelty for the segmentation method or the uncertainty estimates. Our contribution is the translation of these techniques into the clinical domain, where a sanity check contributes to the clinician's demands for fast and informative methods.

---

### Review · AnonReviewer3 · 2018-05-10
**The authors present an uncertainty driven sanity check for brain tumor cavity segmentation.**

**Rating:** 3
**Confidence:** 2

**Review:**


This work presents an uncertainty driven sanity check for brain tumor cavity segmentation. Overall the paper is well written and presented and the results of tumor segmentation is promising. There are however few issues that needs to be addressed;

1. Predicted entropy or the doubt metric is dependent on the quality of segmentation. The metric however does not incorporate any segmentation errors in it. Please explain.

2. Please explain why a 3D CNN was not used. Considering the images are all isotropic after data pre-processing (1x1x1 mm), 3D UNet and DeepMedic has often used for brain lesion and tumor segmentation, the model needs to be compared to those models.



**Special Issue:**

No

---

> ### Comment · ~Alain_Jungo1 · 2018-05-14
> **Response to Reviewer3**
>
> We thank you for your comments about our manuscript. Please find below our clarification to the comments.
>
> 1) It is true that the doubt metric or predicted entropy is dependent on the segmentation quality. However, during test time no expert labels are available. Therefore, we use the doubt metric to assess the segmentation quality.
>
> 2) We favor a slice-wise approach over a volume-wise (3D) approach due to the difference between in-plane resolution (typically around 1x1mm2) and inter-slice spacing (up to 6mm) of the unprocessed images (as recently shown in Baumgartner et al., 2017. An Exploration of 2D and 3D Deep Learning Techniques for Cardiac MR Image Segmentation. https://link.springer.com/chapter/10.1007/978-3-319-75541-0_12). Although resampled to unit voxel resolution, we expect that the network obtains the segmentation crispiness from the good in-plane resolution, which motivates us to process the in-plane separately from the out-planes. Moreover, the 2D approach is fast to compute. Computation times gain in importance when performing Monte Carlo sampling for the uncertainty estimates.
> Additionally, the primary focus of our work is the sanity check, which indeed requires good segmentations but could also be applied to other architectures than the one we used.

---

### Comment · ~Soumya_Ghose1 · 2018-05-05
**The authors present an uncertainty driven sanity check for brain tumor segmentation.**

This work presents an uncertainty driven sanity check for brain tumor segmentation. Overall the paper is well written and presented and the results of tumor segmentation is promising. There are however few issues that needs to be addressed;

1. Predicted entropy or the doubt metric is dependent on the quality of segmentation. The metric however does not incorporate any segmentation errors in it. Please explain.

2. Please explain why a 3D CNN was not used. Considering the 3D UNet and DeepMedic has often used for brain lesion and tumor segmentation, the model needs to be compared to those models.

Recommendation: Neutral

---

> ### Comment · ~Alain_Jungo1 · 2018-05-10
> **Response to Soumya Ghose**
>
> Dear Soumya Ghose,
>
> We thank you for your comments about our manuscript. Please find below our clarification to the comments.
>
> First, we want to clarify that we performed segmentation of brain tumor resection cavities, not brain tumors, as written in the title of your comment.
>
> 1) It is true that the doubt metric or predicted entropy is dependent on the segmentation quality. However, during test time no expert labels are available. Therefore, we use the doubt metric to assess the segmentation quality.
>
> 2) We favor a slice-wise approach over a volume-wise (3D) approach due to the anisotropic resolution (depending on the subject and sequence) of the unprocessed, clinically acquired image volumes, as described in Section 2.2 of the manuscript. Additionally, the primary focus of our work is the sanity check, which indeed requires good segmentations but could also be applied to other architectures than the one we used.

---

### Decision · Program_Chairs · 2018-05-15
**Paper97 Acceptance Decision**

Poster